# Two in One Go: Single-stage Emotion Recognition with Decoupled Subject-context Transformer

## ABSTRACT

Emotion recognition aims to discern the emotional state of subjects within an image, relying on subject-centric and contextual visual cues. Current approaches typically follow a two-stage pipeline: first localize subjects by off-the-shelf detectors, then perform emotion classification through the late fusion of subject and context features. However, the complicated paradigm suffers from disjoint training stages and limited fine-grained interaction between subject-context elements. To address the challenge, we present a single-stage emotion recognition approach, employing a Decoupled Subject-Context Transformer (DSCT), for simultaneous subject localization and emotion classification. Rather than compartmentalizing training stages, we jointly leverage box and emotion signals as supervision to enrich subject-centric feature learning. Furthermore, we introduce DSCT to facilitate interactions between fine-grained subject-context cues in a "decouple-then-fuse" manner. The decoupled query tokens—subject queries and context queries—gradually intertwine across layers within DSCT, during which spatial and semantic relations are exploited and aggregated. We evaluate our single-stage framework on two widely used context-aware emotion recognition datasets, CAER-S and EMOTIC. Our approach surpasses two-stage alternatives with fewer parameter numbers, achieving a 3.39% accuracy improvement and a 6.46% average precision gain on CAER-S and EMOTIC datasets, respectively.

## KEYWORDS

emotion recognition, single-stage framework, and query decouple.

### ACM Reference Format:

Anonymous Author(s). 2018. Two in One Go: Single-stage Emotion Recognition with Decoupled Subject-context Transformer. In *Proceedings of Make sure to enter the correct conference title from your rights confirmation email (Conference acronym 'XX)*. ACM, New York, NY, USA, 10 pages. https://doi.org/XXXXXXX.XXXXXXX

## 1 INTRODUCTION

Automatic human emotion recognition gets increasing research attention in the multimedia community, where studies include inferring emotions from speech [63, 74], image [41, 70] and multi-modalities [36, 38]. Its potential applications span across healthcare, driver surveillance, and diverse human-computer interaction systems [7, 42, 43, 59], reflecting the fundamental role of emotions [10].

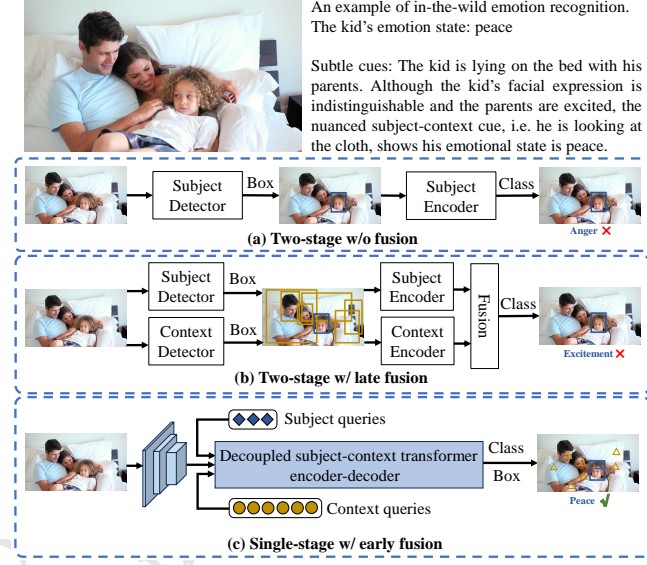

An example of in-the-wild emotion recognition. The kid's emotion state: peace

Subtle cues: The kid is lying on the bed with his parents. Although the kid's facial expression is indistinguishable and the parents are excited, the nuanced subject-context cue, i.e. he is looking at the cloth, shows his emotional state is peace.

**(a) Two-stage w/o fusion**

**(b) Two-stage w/ late fusion**

**(c) Single-stage w/ early fusion**

**Figure 1: Motivation of single-stage framework. Contexts play a vital and nuanced role in emotion recognition. In (a) and (b), prior methods include two stages: subject without or with context (blue and gold rectangles) region localization and emotion classification without or with late fusion. In (c), we propose a single-stage framework for simultaneous localization and classification and decoupled subject-context transformer with early fusion. Our method notices useful and subtle emotional cues (blue and gold triangles).**

In this paper, we focus on the problem of inferring the emotion of one person in a real-world image. Concretely, given an in-the-wild image, we aim to identify the subject's apparent discrete emotion categories (e.g. happy, sad, fearful, or neutral). Existing methods typically involve two stages: subject detection and emotion classification. Conventional approaches primarily emphasize facial cues [6, 47, 56–58, 71, 78], featuring a *two-stage without fusion* paradigm. As depicted in Fig. 1(a), a standard off-the-shelf detector indicates a facial region, and a dedicated face encoder extracts facial features for subsequent classification into distinct emotional categories. Recent advances have increasingly recognized the importance of contextual cues in emotion recognition, like body language, scene semantics, and social interactions [18, 20, 24, 33, 34, 41, 60, 61]. This system is characterized as a *two-stage with late fusion* paradigm. As illustrated in Fig. 1(b), it first identifies subjects and contexts within the image, processes them through independent encoders, and fuses the resulting features for emotion prediction.

While effective, existing approaches are hindered by two primary limitations. Firstly, the disjointed learning processes of emotion classifiers and subject detectors in a two-stage paradigm often result in inefficient computational efficiency. Illustrated in Fig. 2, existing

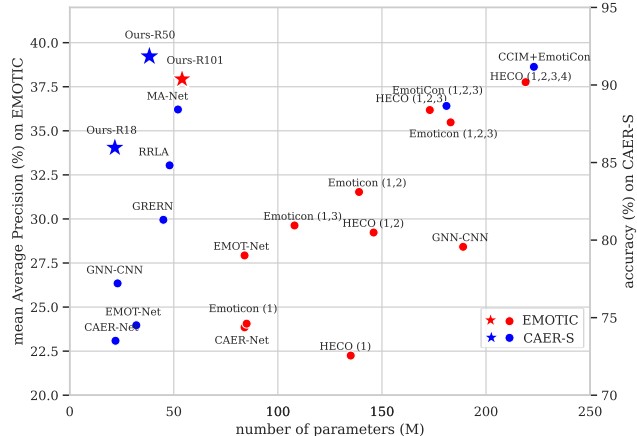

**Figure 2: Performance vs. model efficiency of different methods on EMOTIC (red) and CAER-S (blue). Our proposed single-stage framework (star) achieves state-of-the-art performance with fewer parameters than two-stage prior arts (circle).**

methods' effectiveness is limited with many parameters. Secondly, existing paradigms may exhibit a restricted capacity for subject-context fusion, thereby falling short in addressing real-world images that are susceptible to nuanced contextual influences [32, 55]. Shown in Fig. 1, the first paradigm focuses solely on facial expressions, neglecting essential contextual cues, and the second paradigm's late fusion scheme misses fine-grained subject-context interaction, leading to sub-optimal emotion recognition.

To alleviate the limitation, we introduce a single-stage framework, employing a Decoupled Subject-Context Transformer (DSCT) with early fusion, for simultaneous subject localization and emotion classification, characterized as a *single-stage with early fusion* paradigm. As illustrated in Fig. 1(c), we adopt an encoder-decoder architecture with DSCT, where learnable queries are correlated with the global and multi-scale features for prediction. Rather than disjoint training stages, we jointly leverage box and emotion signals as supervision to enrich subject-centric feature learning, i.e., the framework is trained with a joint loss of classification and localization. Fig. 2 demonstrates that our method is effective and efficient, surpassing two-stage prior arts with fewer parameters.

Furthermore, we introduce DSCT to facilitate interactions between fine-grained subjects and context in a decouple-then-fuse manner. As depicted in Fig. 1(c), the queries are decomposed into subject and context queries to capture the subject's emotional signal, e.g., facial expression, and a wide range of contextual cues, e.g., body posture and gesture, agents, objects, and scene attributes. The decoupled query tokens—subject queries and context queries—gradually intertwine across layers within DSCT. For effective fusion, the spatial and semantic relations between context and subject information are exploited and aggregated. The spatial relation picks up contextual queries with short-range subject-context interaction, such as the subject between objects in hands and close agents. As complementary, the semantic relation chooses contextual queries with long-range subject-context interaction, like the subject between

scene attributes and distant people. Fig. 1(c) shows that the single-stage framework notices useful and subtle emotional cues between the subject and context, e.g. the kid is looking at the father's clothes.

Extensive experiments are conducted on two standard context-aware emotion recognition benchmarks to validate the efficacy of our approach. The proposed framework attains impressive results, achieving 91.81% accuracy on the CAER-S dataset [20] and 37.81% mean average precision on EMOTIC [18]. In the case of similar parameter numbers, the proposal surpasses counterparts by a substantial margin of 3.39% accuracy and 6.46% average precision on CAER-S and EMOTIC respectively. Furthermore, we provide valuable insights by visualizing network output, feature map activation, and query selection, underscoring the proposal can discern useful and nuanced emotional cues of subject and context.

The main contributions can be summarized as follows:

- We present a novel single-stage framework for simultaneous subject localization and emotion classification to address the limitations of disjoint training stages.
- To facilitate fine-grained interactions between subjects and context, we introduce a new decoupled subject-context transformer to decouple and fuse queries across layers.
- The spatial and semantic relations are exploited and aggregated to capture the short-range and long-range subject-context interaction complementarily.
- Extensive experiments and visualization on two standard datasets show that the single-stage framework outperforms two-stage alternatives by a significant margin and excels in capturing useful and nuanced emotional cues.

## 2 RELATED WORK

**Visual Emotion Recognition.** The Visual Emotion Recognition (VER) task can be broadly categorized into two main paradigms. 1) *Two-stage without fusion.* Traditional methods focus on utilizing subject-centric regions while treating contextual areas as noise, as observed in various studies [8, 23, 39, 46, 71]. The pipeline includes subject detection and emotion classification. These studies primarily address challenges associated with label uncertainty [4, 5, 19, 30, 48, 51, 52, 62, 76, 77], micro expressions [37, 67], and disentangled representations [65, 75]. 2) *Two-stage with late fusion.* In recent years, the research has paid increasing attention to context-aware emotion recognition, which emphasizes the use of multiple contexts for more robust emotion classification [18, 20, 24, 34, 35, 49, 60, 61, 73]. In addition to two-stage components, the pipeline includes multi-branch and late fusion characteristics. Typically, a multiple-stream architecture, followed by a fusion network, is employed to independently encode the subject and context information. *Despite the effectiveness of these methods, they suffer from disjoint training stages and limited interaction between fine-grained subject-context elements. In contrast, we present a single-stage approach with an early fusion, employing a Decoupled Subject-Context Transformer, for simultaneous subject localization and emotion classification.*

**End-to-End Object Detection.** The end-to-end framework with vision Transformers stirs up wind in the object detection task. DETR [2] streamlines object detection into one step by a set-based loss and a transformer encoder-decoder architecture. The following works have attempted to eliminate the issue of slow convergence

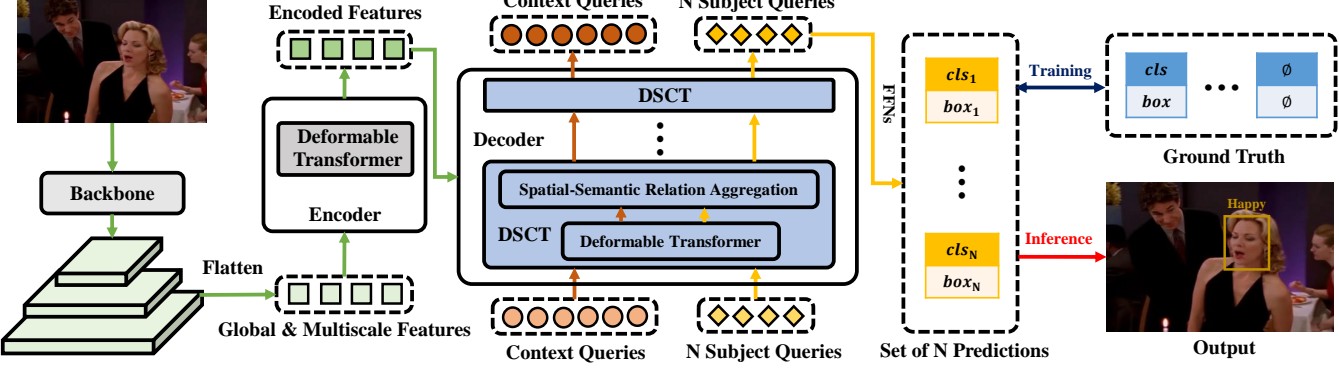

**Figure 3: Overall architecture of our single-stage emotion recognition approach for simultaneous subject localization and emotion classification, employing a Decoupled Subject-Context Transformer (DSCT) with early subject-context fusion.**

by designing architecture [9, 50], query [28, 54, 81], and bipartite matching [3, 21, 22, 68, 69]. The original DETR framework, along with its various adaptations, has not only brought forth a simple yet powerful end-to-end architecture for common object detection but has also been extended to other related tasks, including multiple-object tracking [66], action detection [29], human-object interaction [16, 17], person search [1], and instance segmentation [14, 53]. *We propose the adaptation and modification for VER: First, since we suggest a single-stage framework, we adopt deformable DETR for simultaneous subject localization and emotion classification; Second, as generic objects exhibit distinct and localized characteristics, but contexts are essential and nuanced-related for VER, we introduce a decoupled subject-context transformer to capture contextual interaction.*

## 3 METHOD

### 3.1 Single-stage Framework

Current two-stage approaches for in-the-wild emotion recognition may suffer low efficiency from disjoint training stages and limited interaction between fine-grained subject-context elements. To address the limitation, we introduce a single-stage framework, employing a Decoupled Subject-Context Transformers with early fusion, for subject localization and emotion classification.

**Architecture.** As shown in Fig.3, the system handles an entire image through a CNN backbone, an encoder with deformable transformers, and a decoder with novel Decoupled Subject-Context Transformers (DSCT). Given an image, we extract multi-scale features through the backbone, flatten them in spatial dimensions, and supplement position encoding and level embeddings. The encoder subsequently encodes the global and multi-scale features through six deformable transformers. After that, the decoder correlates the given learnable queries with encoded features with six DSCTs. Finally, the Feed-Forward Networks (FFNs) transform a set of $N$ subject queries into $N$ final predictions, including emotion classes and bounding boxes. We defer to the supplementary material the detailed definition of the architecture, which follows deformable DETR [81]. We jointly train classification and localization to enrich subject-centric feature learning. Furthermore, DSCTs facilitate fine-grained subject-context interactions by early fusion.

**Queries.** Each learnable query is a concatenation of 256-dimension spatial and 256-dimension semantic embeddings. The spatial embedding is decoded into the 2-d normalized coordinate of the reference point and the semantic one into 1) the bounding box as relative offsets w.r.t. the reference point and 2) the corresponding emotion class of the subject. The semantic embeddings of queries adaptively integrate multi-scale image features by sampling locations around the reference points. We refer the reader to the supplementary material for detailed definitions, which follow deformable DETR [81]. We adopt a set of $N$ queries for prediction, where $N$ is typically larger than the average subject number per image in a dataset.

**Optimization.** We use set-level prediction [2] that encapsulates several predictions or ground truth within a set. For clarity, we denote the $i$-th element of a set as $(\text{cls}_i, \text{box}_i)$, where $\text{cls}_i$ represents the categorical emotion label and $\text{box}_i \in [0, 1]^4$ specifies the normalized center coordinate and box's height and width.

During training, since the prediction number is larger than the actual number of subjects in an image, we first pad the set of ground truths with $\emptyset$ to ensure a consistent size. We employ the bipartite matching [2] that computes one-to-one associations between the set of predictions $\hat{y}$ and the padded ground truths $y$:

$$\hat{\sigma} = \arg\min_{\sigma \in \mathfrak{S}_N} \sum_i^N \mathcal{L}_{\text{match}}\left(y_i, \hat{y}_{\sigma(i)}\right), \quad (1)$$

where $\hat{\sigma}$ represents the optimal assignment, $\sigma \in \mathfrak{S}_N$ denotes a permutation of $N$ elements, $\mathcal{L}_{\text{match}}\left(y_i, \hat{y}_{\sigma(i)}\right)$ indicates a pair-wise matching cost between ground truth and a prediction with index $\sigma(i)$. $\mathcal{L}_{\text{match}}$ encompasses a classification loss $\mathcal{L}_{\text{cls}}$ and a box regression loss $\mathcal{L}_{\text{box}}$, expressed as:

$$\mathcal{L}_{\text{match}} = \theta_{\text{cls}}\mathcal{L}_{\text{cls}}(y_i^{\text{cls}}, \hat{y}_{\sigma(i)}^{\text{cls}}) + \theta_{\text{box}}\mathcal{L}_{\text{box}}(y_i^{\text{box}}, \hat{y}_{\sigma(i)}^{\text{box}}), \quad (2)$$

where $\theta_{\text{cls}}, \theta_{\text{box}} \in \mathbb{R}$ are hyperparameters. We efficiently compute the matching results using the Hungarian algorithm [2].

Given the optimal assignment $\hat{\sigma}$, the training loss $\mathcal{L}$ is:

$$\mathcal{L} = \lambda_{\text{cls}}\mathcal{L}_{\text{cls}}(y^{\text{cls}}, \hat{y}_{\hat{\sigma}}^{\text{cls}}) + \lambda_{\text{box}}\mathcal{L}_{\text{box}}(y^{\text{box}}, \hat{y}_{\hat{\sigma}}^{\text{box}}), \quad (3)$$

where $\lambda_{\text{cls}}, \lambda_{\text{box}} \in \mathbb{R}$ are hyperparameters. For matching and training, we employ the focal loss [26] for $\mathcal{L}_{\text{cls}}$ and set $\mathcal{L}_{\text{box}}$ as the $l_1$ loss and generalized IoU loss [45].

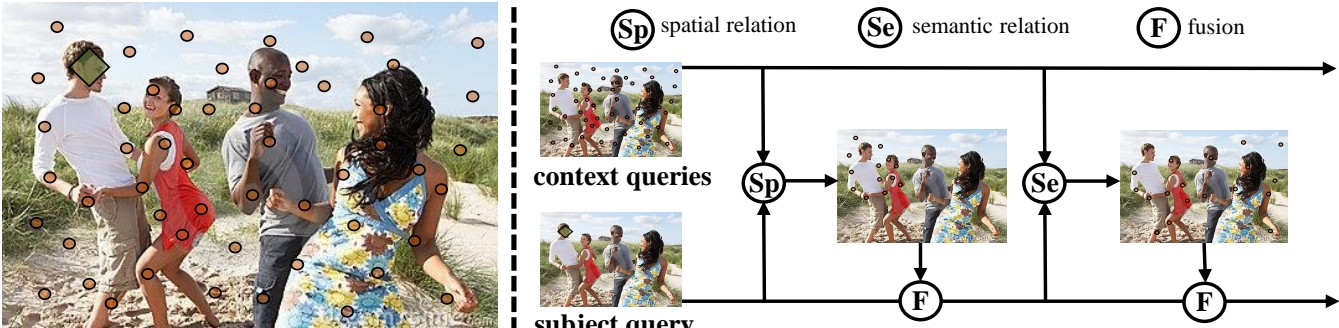

**Figure 4: Illustration of the DSCT. The diamond and circle refer to the subject and context queries. The left figure shows the reference points of the subject and context queries. The right part describes the spatial-semantic relational aggregation.**

During inference, we set the mean of the class output logit as the score of each prediction. For multi-label tasks, subject emotions are determined with a threshold $t$:

$$o = \{i \mid \hat{y}_i > t\}, \tag{4}$$

where $o$ represents the index list of the emotion class. In the case of multi-class tasks, subject emotions are determined as:

$$o = \arg\max_i \{\hat{y}_i\}, \tag{5}$$

where $o$ corresponds to the index of the emotion class.

**Discussion.** Current approaches of emotion recognition usually include two steps of detection and classification, which suffer from low efficiency from disjoint training stages. There, we pursue a single-stage framework to simultaneously recognize the subject's bounding box and emotion class. The deformable DETR pipeline, including the above-mentioned one-stage processing and joint classification and localization loss, aligns well with our demand.

### 3.2 Decoupled Subject-Context Transformer

To facilitate interactions between fine-grained subject and contextual elements, we introduce a novel Decoupled Subject-Context Transformer (DSCT), which treats queries in a "decouple-then-fuse" manner and exploits spatial-semantic relational aggregation.

**Decouple then Fuse.** Before the DSCT, the queries are decomposed into subject and context queries. As shown in Fig. 3, we adopt $N$ subject queries and a large number of context queries as input queries of the decoder, where all queries have the same tensor size. In DSCT, both types of queries are correlated with multi-scale image features through the base deformable transformer, and then subject queries integrate context queries by spatial-sentimental relational aggregation before output. As illustrated in the left section of Fig. 4, the reference point of the subject query primarily attends to the subject area to capture the subject's emotional signal, e.g., facial expression, while the reference points of the context queries are distributed across the entire image to pick up extensive and subtle contextual cues, e.g., body posture and gesture, surrounding agents, and scene attributes like grass and sky.

**Spatial-Semantic Relational Aggregation.** As shown in the right part of Fig. 4, the DSCT fuses context queries based on their spatial-semantic relationships w.r.t. the subject query.

The DSCT first picks up the short-range contextual cues, such as objects in hands and close agents, based on the relative spatial distance between the subject and context queries. For each context query, the distance is calculated as the Euclidean distance between reference points of the context and subject queries. For the subject and context queries, we denote their coordinate vectors of the reference points as $p_S^n$, where $n = 1, ..., N$ and $N$ is the total number of the subject queries, and $p_C^m$, where $m = 1, ..., M$ and $M$ is the total number of the context queries. The relative spatial distance for a pair of subject and context query $d_m^n$ is computed as:

$$d_m^n = \|p_C^m - p_S^n\|_2. \tag{6}$$

Then we select $K_{sp}$ queries of the shortest spatial distance from total $M$ context queries for each subject query.

As complementary, the long-range contextual signals, like scene attributes and distant people, are chosen via the semantic relevance of the subject and context queries. For each context query, the relevance is calculated as the similarity between semantic embeddings of context and subject queries. For the subject and context queries, we denote their semantic embeddings as $E_S^n$, where $n = 1, ..., N$ and $N$ is the total number of the subject queries, and $E_C^m$, where $m = 1, ..., M$ and $M$ is the total number of the context queries. The semantic relevance for a pair of subject and context query $r_m^n$ is calculated as:

$$r_m^n = \mathrm{dot}(E_C^m, E_S^n). \tag{7}$$

Since the semantic embeddings are processed with image features in the same architecture, we can measure their similarity without the transforming matrices in previous methods [24, 61]. Then we select $K_{sm}$ queries of the smallest semantic relevance from total $M$ context queries for each subject query.

Finally, we adopt relevance re-weighting fusion to integrate the context queries into the subject query. We denote their semantic embeddings as $E_S$ and $E_C^k$, where $k = 1, ..., K_{sm} + K_{sp}$. The attention weight $w^k$ is computed by the dot product of $E_S$ and $E_C^k$. Then the softmax function makes the sum of attention weights to be 1. After that, the fused contextual subject query $\hat{E}_S$ is defined as:

$$\hat{E}_S = \sum_{k=1}^{K_{sm}+K_{sp}} w^k E_C^k + E_S. \tag{8}$$

**Discussion.** In the object detection task, queries can capture effective information from distinctive and localized areas. While in

the task of emotion recognition, contextual cues are essential and nuanced-related, we introduce the novel DSCT to capture sufficient and useful contexts based on spatial-semantic relationships.

## 4 EXPERIMENTS

### 4.1 Implementations

We set the number of queries $N$ to 4 and 9 for CAER-S [20] and EMOTIC [18]. We set $N$ to 4 and 9. To facilitate the training, we initialize the weights of architecture and borrow 300 context queries from Deformable DETR [81], which was pre-trained on COCO [27]. Our batch size is 32, and we set hyperparameters $\theta_{box}$, $\lambda_{box}$, $\theta_{cls}$, and $\lambda_{cls}$ to 5, 5, 2, and 5, respectively. For evaluation, we first use non-max-suppression to remove the duplicate subjects and then select the subject that exhibits the highest bounding box overlap with the ground truth. The experiments were conducted using 8 GPUs of the NVIDIA Tesla A6000. We show details about architectural configurations, training strategies, and preprocessing steps, which follow those outlined in [81], in the supplementary material.

### 4.2 Datasets

We conducted extensive experiments on two typical and popular context-aware emotion recognition datasets in real-world scenarios, namely the CAER-S [20] and EMOTIC [18].

The CAER-S dataset consists of 70,000 images, randomly divided into training (70%), validation (10%), and testing (20%) sets. Annotations include face bounding boxes and multi-class emotion labels. The dataset encompasses seven emotion categories: Surprise, Fear, Disgust, Happiness, Sadness, Anger, and Neutral. Performance on this dataset is measured using overall accuracy (acc) [20].

The EMOTIC dataset [18] contains a total number of 23,571 images and 34,320 annotated agents, which are randomly split into training (70%), validation (10%), and testing (20%) sets. Annotations include body and head bounding boxes, as well as multi-label emotion categories. EMOTIC encompasses 26 emotion categories: Affection, Anger, Annoyance, Anticipation, Aversion, Confidence, Disapproval, Disconnection, Disquietment, Doubt/Confusion, Embarrassment, Engagement, Esteem, Excitement, Fatigue, Fear, Happiness, Pain, Peace, Pleasure, Sadness, Sensitivity, Suffering, Surprise, Sympathy, Yearning. Performance on EMOTIC is evaluated based on the mean Average Precision (mAP) for all classes [18].

### 4.3 Quantitative and Qualitative Results

The performance of various methods on CAER-S and EMOTIC datasets is presented in Table 1 and Table 2. To facilitate a fair comparison, we categorize the methods into two groups based on the number of parameters: similar-parameter measures and larger-parameter ones, using a threshold of 100 Million (M) parameters. For the methods without released code, we count their parameters through their backbone configuration in paper and present the details in the rightmost column. Subscripts in EmotiCon [34] and HECO [61] correspond to specific context modalities as mentioned in their respective papers. The performance of the methods is sourced from their original papers or re-implemented results of other papers. Our proposal framework outperforms similar-parameter methods by a notable margin, achieving a significant 3.39% improvement on CAER-S and an impressive 6.46% boost on

| Methods | Acc (%) | Param.(M) | Backbone |
|---|---|---|---|
| *With <100M parameters* | | | |
| Ours-R18 | 84.96 | 22 | ResNet18 |
| CAER-Net-S [20] | 73.51 | 22 | 12-layer CNN |
| GNN-CNN [72] | 77.21 | 23 | VGG16 |
| EfficientFace [80] | 85.87 | 25 | MobileNet28, ResNet18 |
| EMOT-Net [18] | 74.51 | 32 | ResNet18 × 2 |
| SIB-Net [25] | 74.56 | 33 | ResNet18 × 3 |
| **Ours-R50** | **91.81** | **39** | ResNet50 |
| GRERN [11] | 81.31 | 45 | ResNet101 |
| RRLA [24] | 84.82 | 48 | ResNet50, RCNN50 |
| MA-Net [79] | 88.42 | 52 | Multi-Scale ResNet18 |
| *With >100M parameters* | | | |
| EmotiCon [34] | 88.65 | 181 | OpenPose, RobustTP, Megadepth |
| VRD [13] | 90.49 | 380 | {VGG19, ResNet50, FRCNN50} × 2 |
| CCIM+EmotiCon [60] | 91.17 | 223 | OpenPose, RobustTP, Megadepth, ResNet101 |

**Table 1: Performance and model efficiency on the CAER-S.**

EMOTIC. Notably, the proposal even surpasses larger-parameter approaches, underscoring its suitability for emotion recognition when compared to two-stage methods.

We present the qualitative results in Fig. 5 and Fig. 6, depicting the bounding boxes and emotion classes output by our proposal framework, alongside those produced by the EMO-Net [18], a representative two-stage late-fusion method. To enhance the clarity of the proposal's output, we include visual indicators for subject queries' reference points (colored in red) and sampling locations. Outputs of different subjects are color-coded for differentiation. These visualizations illustrate that the proposal consistently yields high-quality results, showcasing superior classification accuracy when compared to EMO-Net. In EMOTIC, the EMO-Net might neglect subtle emotions like "Pain", or produce wrong even opposite emotions like "Disapproval". In CAER-S, when the facial expression is not distinguishing, the EMO-Net is confused with "Anger" and "Neutral", "Happy" and "Surprise", or "Surprise" and "Fear".

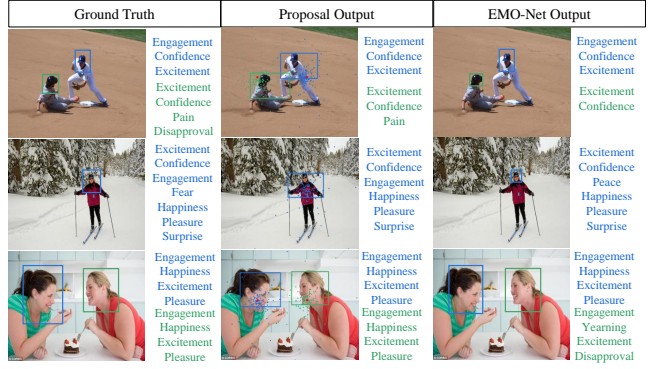

**Figure 5: The output visualization on EMOTIC.**

| Methods | mAP (%) | Param.(M) | Backbone |
|---|---|---|---|
| *With <100M parameters* | | | |
| Ours-R50 | 37.26 | 39 | ResNet50 |
| **Ours-R101** | **37.81** | 58 | ResNet101 |
| EMOT-Net [18] | 27.93 | 84 | YOLO, ResNet18 |
| CAER-Net [18] | 20.84 | 84 | YOLO, CNN-12 |
| EmotiCon(1) [34] | 31.35 | 85 | OpenPose, 15-layer CNN |
| *With >100M parameters* | | | |
| EmotiCon(1,3) [34] | 35.28 | 108 | OpenPose, 15-layer CNN, Medadepth |
| HECO(1) [61] | 22.25 | 135 | YOLO, Alphapose, ResNet18 |
| EmotiCon(1,2) [34] | 32.03 | 139 | OpenPose, RobustTP, ResNet18 |
| HECO(1,2) [61] | 36.18 | 146 | YOLO, Alphapose, ResNet18 × 2 |
| HECO(1,2,3) [61] | 34.93 | 173 | YOLO, Alphapose, ResNet50, ResNet18 × 2 |
| EmotiCon(1,2,3) [34] | 32.03 | 183 | OpenPose, RobustTP, ResNet18, Megadepth |
| GCN-CNN [72] | 28.16 | 189 | YOLO, VGG16, 6-layer GCN |
| HECO(1,2,3,4) [61] | 37.76 | 219 | YOLO, Alphapose, {ResNet18, ResNet50} ×2, Faster RCNN50 |
| EmotionCLIP [73] | 32.91 | 577 | YOLO, ViT_b_32 |

**Table 2: Performance and model efficiency on the EMOTIC.**

| Input | Ground Truth | Proposal Output | EMO-Net Output |
|---|---|---|---|

**Figure 6: The output visualization on CAER-S.**

## 4.4 Visualization and Analysis

**Classification and localization.** We conducted experiments to fine-tune $\theta_{cls}$ and $\lambda_{cls}$ on the EMOTIC dataset [18] and keep $\theta_{box} = 5$ and $\lambda_{box} = 5$. The results of these hyperparameter experiments are thoughtfully presented in Table 3. Notably, the most compelling performance is achieved when $\theta_{cls} : \theta_{box}$ is set to 2:5, and $\lambda_{cls} : \lambda_{box}$ is set to 5:5. We can see adding appropriate localization loss can boost classification performance and facilitate subject-centric feature learning. Besides, we noticed that $\theta_{cls}$ has minimal impact on performance, whereas $\lambda_{cls}$ significantly influences the results.

**Feature map activation.** We visualize feature map activation of methods of different paradigms. We select EMO-Net [18] as a

| $\theta_{cls}$ | 5 | 10 | 15 | 2 | 2 | 2 |
|---|---|---|---|---|---|---|
| $\lambda_{cls}$ | 2 | 2 | 2 | 5 | 10 | 15 |
| mAP (%) | 35.41 | 35.89 | 35.41 | **36.01** | 34.99 | 34.64 |
| $\theta_{cls}$ | 2 | 5 | 10 | 15 | 1 | 1 |
| $\lambda_{cls}$ | 2 | 5 | 10 | 15 | 10 | 8 |
| mAP (%) | 35.94 | 35.00 | 35.05 | 34.75 | 34.70 | 35.45 |

**Table 3: Performance of different classification coefficients.**

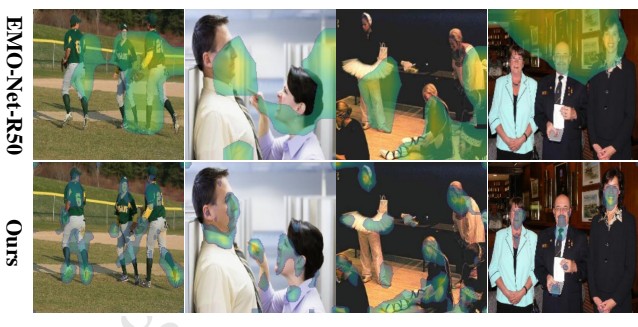

**Figure 7: The visualization of feature maps activation.**

representative two-stage method. For fairness, we re-implement EMO-Net using ResNet50 as the backbone to achieve a similar model complexity to the proposal (referred to as EMO-Net-R50). The selected feature maps originate from the final layer of the ResNet50 backbone. In Fig. 7, a clear distinction emerges: EMO-Net-R50 exhibits a tendency to emphasize a few large regions, while the proposal consistently places importance on smaller, more intricate areas. This observation suggests that the proposal excels in the precise handling of fine-grained subject-context cues compared to conventional two-stage methods of coarse-grained cues.

**Feature sampling positions of queries** We visualize the reference point and feature sampling positions of the normal subject queries and contextual subject queries of the DSCT in Fig. 8. The reference points are drawn as big blue stars and the sampling positions as small grey circles. For the normal subject query, the sampling positions only rely on its own sampling points [81]. For the contextual subject query, the sampling positions rely on both the sampling points of itself and context queries. We can see the sampling positions of normal subject queries only cover the subject area while the contextual ones are densely distributed across the image. The quantitative result shows the contextual query of DSCT outperforms the normal one with a 0.71% precision improvement. The gain can be attributed to aggregating extensive contextual cues, which are essential for in-the-wild emotion recognition.

**Multiple subjects.** We conduct an evaluation on images with varying subject numbers. We select EMO-Net-R50 and EMO-Net-R50-M, which further masks subjects in the context [20], for comparison. Table 4 presents the performance on EMOTIC for images with different subject numbers. As the subject number in an image increases, the complexity of subject-context interaction also rises. Notably, the proposal maintains stable performance with increasing subject numbers, while EMO-Net-R50's performance deteriorates. This observation verifies our early fusion proposal can handle complex interactions better than late fusion two-stage methods.

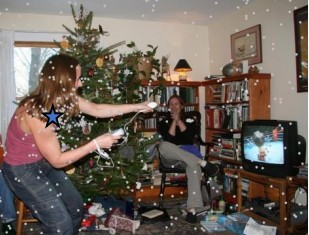

**Figure 8: The visualization of the reference point (blue star) and sampling positions (grey circle) of queries.**

| Subject # | 1 | 2 | 3 | 4 | >=5 |
|---|---|---|---|---|---|
| Image # | 2444 | 938 | 234 | 37 | 29 |
| EMO-Net-R50 | 22.34 | 20.50 | 19.62 | 18.77 | 18.06 |
| EMO-Net-R50-M | 22.54 | 20.96 | 19.65 | 19.20 | 19.50 |
| Ours-R50 | 36.91 | 35.20 | 31.20 | 40.96 | 35.97 |

**Table 4: Performance on images with multiple subjects.**

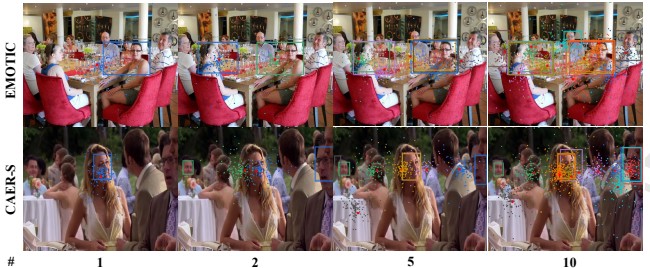

**Figure 9: The position visualization of subject queries.**

| Set Number | 1 | 2 | 3 | 4 | 5 |
|---|---|---|---|---|---|
| EMOTIC (mAP %) | 37.14 | 36.71 | 36.61 | **37.26** | 36.70 |
| CAER-S (Acc %) | 91.78 | 91.57 | 91.39 | 91.57 | 91.47 |
| Set Number | 6 | 7 | 8 | 9 | 10 |
| EMOTIC (mAP %) | 36.97 | 36.03 | 36.26 | 35.92 | 35.68 |
| CAER-S (Acc %) | 91.52 | 91.59 | 91.42 | **91.81** | 91.44 |

**Table 5: Ablation study on subject query number.**

## 4.5 Ablation Study

**Subject Query Number.** We conduct experiments to investigate the impact of query number setting. Table 5 presents the performance of different query numbers. Fig. 9 offers a visualization of bounding boxes (colorful rectangles), reference points (red circles), and sampling locations (colorful circles) corresponding to different subject query numbers. Table 5 shows the proposal achieves the best result on EMOTIC and CAER-S when the query number is 4 and 9 respectively. Notably, we observe a consistent performance stability trend as the query number increases, ranging from 1 to 6 on EMOTIC and from 1 to 10 on CAER-S. Fig. 9 also indicates that different subject queries attend to separate subject areas.

**Components of DSCT.** We assess the impact of components of DSCT on the EMOTIC dataset [18], and the results are displayed

| Baseline | Decouple-fuse | Spatial | Semantic | mAP (%) |
|---|---|---|---|---|
| ✓ | ✗ | ✗ | ✗ | 36.55 |
| ✓ | ✓ | ✗ | ✗ | 36.73 |
| ✓ | ✓ | ✓ | ✗ | 37.11 |
| ✓ | ✓ | ✗ | ✓ | 36.85 |
| ✓ | ✓ | ✓ | ✓ | **37.26** |

**Table 6: Ablation study on DSCT components.**

in Table 6. The categories include "baseline" (only subject queries), "Decouple-fuse" (decouple queries and then fuse), "Spatial" (select $K_{sp}$ context queries with shortest spatial distance), and "Semantic" (select $K_{se}$ context queries with semantic relevance). As we can see, the DSCT enhances performance by 0.71%, highlighting the importance of sufficient contextual interaction and fusion. Specifically, decoupling and fusing context queries improve performance by 0.18%, and selecting context queries based on spatial and semantic relation boosts the result by an extra 0.56% and 0.12%. The results demonstrate the effectiveness of each proposed component.

**Selection of Spatial and Semantic Relation.** We conduct experiments with varying values of $K_{sp}$ and $K_{se}$ on EMOTIC [18] to evaluate the sensitivity of Spatial and Semantic Relation parameters. As shown in Figure 11, the optimal performance is achieved when $K_{sp}$ is set to 100. The best performance is 0.41% higher than the one when $K_{sp}$ is 300. This observation suggests that not all contextual information is valuable for effective emotion recognition. The selection of the 100 closest context queries w.r.t. the subject query also aligns with gradual interaction decay in [61]. Figure 10 shows spatial relational selection keeps short-range contextual cues such as objects in hands and close agents. Besides, the optimal performance is achieved when $K_{se}$ is set to 50, which is 0.39% higher than the one when $K_{se}$ is 300. This suggests that some contextual information is a disturbance. Figure 10 shows semantic relational selection preserves long-range contextual signals, like scene attributes and distant people. We can see two selections are complementary.

**Re-weighting Strategy.** We conduct experiments of re-weighting fusion strategies for context queries on the EMOTIC dataset. The results, shown in Table 7, indicate the performance under different re-weighting strategies: Semantic (semantic relevance weights based on semantic relation w.r.t the subject query), Equal (equal weights), Spatial (spatial distance weights based on spatial relation w.r.t the subject query), and Attentive (attentive weights learned from the subject query through a linear layer). The findings reveal that the best performance is achieved when employing semantic re-weighting, which performs better than equal re-weighting by 0.53%. The observation aligns well with previous studies that contexts have variant contributions to emotion recognition [24, 61].

| Re-weighting strategy | Semantic | Equal | Spatial | Attentive |
|---|---|---|---|---|
| mAP % | **37.26** | 36.73 | 36.48 | 36.93 |

**Table 7: The ablation study on re-weighting strategy.**

**Fusion location.** We conduct experiments of different locations for subject-context fusion on the EMOTIC dataset and show the results in Table 8. The decoder has six layers, and we perform the query fusion in different layers. The best performance is achieved

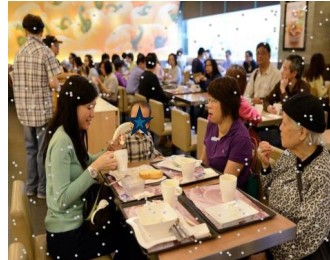
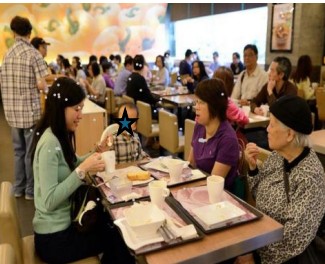
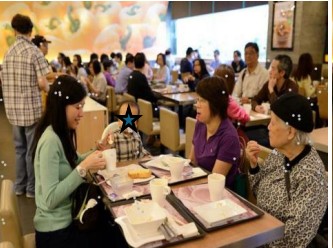
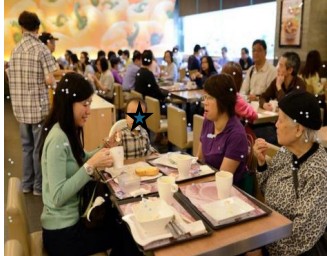

| Decouple-fuse | Spatial | Semantic | Spatial-Semantic |

Figure 10: The position visualization of the subject query (blue star) and selected context queries (grey spots).

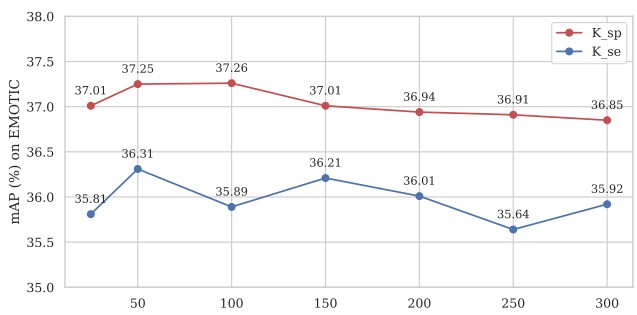

Figure 11: The evaluation of $K_{sp}$ and $K_{se}$ on EMOTIC [18]

when fusing in the 1st to 6th layers, which exceeds fusing only in the 6th layer by 0.57%. It can be explained by that early fusion can capture fine-grained subject-context interaction to boost performance. The observation aligns with the psychological studies [32, 55].

| locations | 1st to 6th | 2nd to 6th | 4th to 6th | 6th |
|---|---|---|---|---|
| mAP % | **37.01** | 36.58 | 36.82 | 36.44 |

Table 8: The ablation study on fusion location.

**Feature Extractor.** To evaluate the influence of feature extractors, we conducted experiments with various backbone architectures for the proposal on both EMOTIC and CAER-S datasets. The results, as depicted in Table 9, include performance metrics and corresponding parameter counts. Specifically, "R" refers to ResNet [12], and "WR" designates Wide ResNet [64]. For ResNet50, we utilized a pre-trained backbone from deformable DETR as the initialization. The optimal performance on EMOTIC is attained when using ResNet-101 as the backbone, while on CAER-S, ResNet-50 yields the best results. Interestingly, it's evident that the relationship between performance and parameter counts is not linear on both datasets. Moreover, the performance on CAER-S seems to be significantly influenced by the choice of pre-trained initialization.

**DETR-like Architecture.** The family of DETR-like architectures has gained significant momentum in the object detection task. To assess the impact of incorporating different DETR-like architectures into proposal, we conducted experiments on the CAER-S dataset by leveraging the detrex platform [44]. For equitable comparisons, all architectures utilize ResNet50 as the backbone. The results, summarized in Table 10, reveal the performance of proposal

| Backbone | Param. (M) | EMOTIC (mAP %) | CAER-S (Acc %) |
|---|---|---|---|
| R18 | 22 | 35.68 | 84.98 |
| R34 | 33 | 34.40 | 84.52 |
| R50 | 39 | 37.26 | **91.81** |
| R101 | 58 | **37.81** | 85.39 |
| R152 | 74 | 37.32 | 83.88 |
| WR50 | 82 | 34.46 | 85.71 |
| WR101 | 140 | 34.07 | 83.16 |

Table 9: Ablation study on feature extractor.

with various DETR-like architectures. Notably, there is a performance gap of around 1% between DETR-based and deformable DETR-based architectures. However, performances remain comparable within DETR-based and deformable DETR-based architectures even though they adopt different techniques.

| Architecture | Acc % | Architecture | Acc % |
|---|---|---|---|
| DETR [2] | 89.74 | DINO [69] | 89.27 |
| Anchor-DETR [54] | 90.42 | Deformable DETR [81] | 91.81 |
| DAB-DETR [9] | 87.49 | DAB-D-DETR [9] | 91.39 |
| DN-DETR [21] | 89.79 | H-D-DETR [15] | 91.65 |
| Conditional-DETR [31] | 90.07 | DETA [40] | 91.77 |

Table 10: Ablation study on different DETR architectures.

## 5 CONCLUSION

This paper introduces a single-stage visual emotion recognition framework with Decoupled Subject-Context Transformers (DSCT). The proposal predicts subjects' emotions and locations simultaneously and processes subject and context emotional cues by early fusion. We evaluate our single-stage framework on two widely used context-aware emotion recognition datasets, CAER-S and EMOTIC. Our approach surpasses two-stage alternatives with fewer parameter numbers, achieving a 3.39% accuracy improvement and a 6.46% average precision gain on CAER-S and EMOTIC datasets, respectively. We observe that the joint training of localization and classification can facilitate subject-centric feature learning. Besides, we find that early fusion improves handling the fine-grained subject-context interaction, e.g. multiple subjects in one scene. We also explore the spatial and semantic relationships between subject and contextual cues for more effective interaction and fusion.

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
