# OpenReview forum: "Two in One Go: Single-stage Emotion Recognition with Decoupled Subject-context Transformer"
_acmmm.org/ACMMM/2024/Conference — MM2024 Poster_

### Official Review · Reviewer_efyW · 2024-05-17

**Rating:** 2
**Confidence:** 4

**Summary:**

The authors propose a DSCT to facilitate interactions between fine-grained subject and contextual elements. They think this two semantic has disjoint training and insufficient interaction. They jointly leverage box and emotion signals as supervision to enrich subject-centric feature learning.

**Strengths:**

1. The proposed method has relatively few parameters, yet good performance.This reduces the complexity and inefficiency associated with the traditional two-stage pipeline.
2. DSCT enhances the interaction between subject and context cues by decoupling them into separate queries and then fusing them through multiple layers. This allows the model to capture fine-grained details from both the subject and the surrounding context.

**Limitations:**

1. The author uses two different semantic queries for their tasks. How to decompose the queries into subject and context.
2. The paper primarily compares the proposed method with two groups based on the number of parameters: similar-parameter measures and larger-parameter ones, using a threshold of 100 Million (M) parameters. However, there is no related single-stage extensively compared or no intuitive contrast with other recent single-stage or end-to-end emotion recognition models to verify the effectiveness of the proposed single-stage query approach. Please add a comparison to 2023 related papers.
3. This paper proposes contextual semantics and queries within the image but does not have an intuitive consideration of how it handles long-range contextual signals, such as emotion variations or situational factors that might influence emotions.

**Suitability:**

2

---

### Official Review · Reviewer_6DJA · 2024-05-23

**Rating:** 4
**Confidence:** 3

**Summary:**

The authors propose a single-stage framework using a Decoupled Subject-Context Transformer (DSCT) that integrates subject localization and emotion classification simultaneously. This approach enriches subject-centric feature learning by using both box and emotion signals as supervision and enables detailed interaction between subject and context cues through a "decouple-then-fuse" mechanism. Extensive experiments on the CAER-S and EMOTIC datasets demonstrate the proposed method's effectiveness, achieving higher accuracy and precision compared to existing two-stage methods while using fewer parameters.

**Strengths:**

1. The method seems reasonable and the results are good.

2. The idea of 'decouple-then-fuse' is interesting.

**Limitations:**

1.As the author claimed that one of the limitations of the two-stage methods is that they ‘often result in inefficient computational efficiency’, which is caused by the disjointed learning processes of emotion classifiers and subject detectors. Is there any experimental result that can support that the proposed one-stage method is more efficient than the two-stage methods? I did not find corresponding results. Or giving some theoretical analysis is also good.

2.There are some hyper parameters in the loss function. It would be better to introduce how the choose the value of them.

3.I am wondering what the ‘decouple’ operation is to disentangle the context-subject content. All I can see is to use different queries to get them from a Transformer. Is this right?

**Suitability:**

2

---

### Official Review · Reviewer_h8Lg · 2024-05-25

**Rating:** 5
**Confidence:** 4

**Summary:**

The paper proposes a single-stage emotion recognition transformer. In conjunction with emotion recognition, the proposed framework predicts the subject’s localisation (bounding box) with no additional stages. This is done through the simultaneous processing of the subject’s localisation and emotion recognition by employing a decoupled subject-context transformer. The method is evaluated on the CAER-S and EMOTIC datasets.

**Strengths:**

The paper is well written and clear. The method is novel for the emotion recognition task, with the minimal preprocessing requirement. The experiments are done on two challenging in-the-wild datasets. Considering that there are <40M training parameters, the results look impressive.

**Limitations:**

1- One of the primary concerns is with consideration of spatial-semantic relationships with respect to the subject query. This is because the paper makes the naive assumption that the relative distance between subject and context is directly proportional to subject-context relevance. This is not always true as the images considered are the 2D ‘projection’ of the 3D perceived world. As a result of the assumption, the short-range contextual cues are naively picked as the few shortest spatial Euclidean distance from the image coordinates. However, the long-range contextual signals involve semantic relevance.
2- The other concern/query is if the context queries are ‘borrowed’ from Deformable DETR[81] before the training, is the proposed method truly a single-stage then?
3- L154: How are learnable queries correlated with the global and multi-scale features. Is the conclusion derived from the qualitative results (for example, Figure 6, Figure 8, etc.)?
4- L399: How are reference of the context queries distributed? Uniformly across the image?  Is this what ‘Sampling Locations’ in the Supplementary results about?
5- How is $K_{sp} $ and $K_{sm}$ chosen?
6- Also, request the authors to see other existing methods (for example, [1]) which are related to the current work of jointly predicting emotion classes and face localisation.
[1] "Estimation of continuous valence and arousal levels from faces in naturalistic conditions", Antoine Toisoul, Jean Kossaifi, Adrian Bulat, Georgios Tzimiropoulos and Maja Pantic, published in Nature Machine Intelligence, January 2021

**Suitability:**

3

---

### Official Review · Reviewer_t6Jw · 2024-05-28

**Rating:** 4
**Confidence:** 3

**Summary:**

This paper distinguishes itself from traditional two-stage methods by proposing a novel one-stage approach for emotion recognition in context. The authors introduce a single-stage framework that effectively reduces the number of model parameters while achieving superior performance.

**Strengths:**

This innovative approach integrates the tasks of subject localization and emotion classification into a unified process, addressing the inefficiencies of disjointed training stages found in previous two-stage models. The reduction in parameters not only enhances computational efficiency but also demonstrates significant improvements in accuracy and precision on benchmark datasets, underscoring the potential of the proposed method for real-world applications.

**Limitations:**

Although the motivation of the paper is strong, I still do not fully understand how the context query and subject query ensure the theoretical effectiveness claimed by the authors. For instance, if there are multiple subjects within a single image exhibiting conflicting emotional cues, it is unclear how the model can accurately resolve these conflicts and make correct judgments. The manuscript lacks detailed explanations and empirical evidence demonstrating how these queries interact and contribute to the overall performance improvements. Further clarification and more comprehensive experimental results are needed to validate the theoretical claims and to provide a clearer understanding of the mechanism behind the proposed approach, particularly in scenarios with conflicting emotional signals.

**Suitability:**

2

---

### Meta-Review · Senior_Area_Chairs · 2024-07-10

**Recommendation:** Accept (Poster)
**Confidence:** 5

**Metareview:**

This manuscript received overall positive ratings. The reviewers kept the positive evaluation of the paper after the rebuttal.

The only negative point of the paper is that it is not multi-modal, since a single modality is used.